# A novel twelve class fluctuation test reveals higher than expected mutation rates for influenza A viruses

Matthew D Pauly[1], Megan C Procario[1], Adam S Lauring[1,2]*

[1]Department of Microbiology and Immunology, University of Michigan, Ann Arbor, United States; [2]Division of Infectious Diseases, Department of Internal Medicine, University of Michigan, Ann Arbor, United States

**Abstract** Influenza virus' low replicative fidelity contributes to its capacity for rapid evolution. Clonal sequencing and fluctuation tests have suggested that the influenza virus mutation rate is $2.7 \times 10^{-6}$ - $3.0 \times 10^{-5}$ substitutions per nucleotide per strand copied (s/n/r). However, sequencing assays are biased toward mutations with minimal fitness impacts and fluctuation tests typically investigate only a subset of all possible single nucleotide mutations. We developed a fluctuation test based on reversion to fluorescence in a set of virally encoded mutant green fluorescent proteins, which allowed us to measure the rates of selectively neutral mutations representative of the twelve different mutation types. We measured an overall mutation rate of $1.8 \times 10^{-4}$ s/n/r for PR8 (H1N1) and $2.5 \times 10^{-4}$ s/n/r for Hong Kong 2014 (H3N2) and a transitional bias of 2.7–3.6. Our data suggest that each replicated genome will have an average of 2–3 mutations and highlight the importance of mutational load in influenza virus evolution.

*For correspondence: alauring@ med.umich.edu

**Competing interests:** The authors declare that no competing interests exist.

## Introduction

The rapid evolution of influenza virus has led to reduced vaccine efficacy, widespread drug resistance, and the annual emergence of novel strains. While complex ecological, environmental, and host demographic factors influence the evolutionary dynamics of influenza virus, the virus' adaptability is driven in large part by its capacity to generate genetic diversity through mutation and reassortment (*Nelson and Holmes, 2007*). Like other RNA viruses, influenza virus replicates with extremely low fidelity. The influenza virus RNA-dependent RNA polymerase (RdRp) complex, which includes the viral proteins PB1, PB2, PA, and NP, lacks proofreading and repair activity (*Te Velthuis and Fodor, 2016*). Its mutation rate has been reported to be approximately $10^{-5}$ to $10^{-6}$ mutations per nucleotide per cellular infection (*Sanjuán et al., 2010*; *Suárez-López and Ortín, 1994*; *Suárez et al., 1992*; *Nobusawa and Sato, 2006*; *Parvin et al., 1986*; *Bloom, 2014*).

An accurate accounting of influenza virus' mutation rate and mutational bias is essential for defining its evolutionary dynamics and for informing control efforts. The mutation rate will determine the probability that a mutation conferring drug resistance, antibody escape, or broadened host range will be generated within a given virus population. It will also define a virus' sensitivity to drug-induced lethal mutagenesis, a broad-spectrum antiviral strategy that exploits the high mutation rate and low mutational tolerance of many RNA viruses (*Anderson et al., 2004*; *Bull et al., 2007*). We have shown that the antiviral activity of three different nucleoside analogues is due to increased viral mutation rates, and a new anti-influenza drug, favipiravir, has been found to act through a similar mechanism (*Cheung et al., 2014*; *Pauly and Lauring, 2015*; *Baranovich et al., 2013*). As in other RNA viruses, the mutational bias of the influenza polymerase complex is largely undefined. While viral mutation rates are typically reported as a single measurement, each of the 12 distinct

nucleotide substitutions (for example, A to C, G to U) will have its own rate. The rates of these mutational classes will in turn determine the accessibility of various nucleotide and amino acid substitutions. Many RNA viruses appear to exhibit a pronounced bias toward transition mutations (*Sanjuán et al., 2010*). Because transitions are more likely to be synonymous than transversions, this bias can impact models of molecular evolution and inferences of natural selection based on dN/dS ratios.

Mutations are typically reported as either frequencies or rates (*Sanjuán et al., 2010*; *Belshaw et al., 2011*). Mutation frequency is the number of mutations identified in a sample per nucleotide sequenced. Frequency measurements therefore quantify not only the rate at which a mutation is generated but also that mutation's ability to persist in a population. In contrast, mutation rates measure how many mutations are made in a discrete unit of time (for example, per infection cycle or strand copied) and are a better representation of polymerase error. Viral mutation rates have often been measured by Sanger sequencing of randomly selected clones obtained through plaque purification or limiting dilutions (*Tromas and Elena, 2010*; *Vignuzzi et al., 2006*; *Eckerle et al., 2007*; *Nobusawa and Sato, 2006*; *Parvin et al., 1986*; *Bloom, 2014*). Mutation frequencies obtained in this manner can be converted to mutation rates by adjusting for the number of replication cycles prior to sampling (*Sanjuán et al., 2010*). With these adjustments, sequencing-based estimates of influenza virus mutation rates range from $7.1 \times 10^{-6}$ to $4.5 \times 10^{-5}$ substitutions per nucleotide per cell infection cycle. While sequencing approaches can potentially measure the rate of all mutational classes, they lack precision and have poor power for detecting differences across strains or conditions. They are also biased towards sampling of genomes with higher fitness. Next generation sequencing platforms have increased the throughput and power of clonal sequencing, but in many cases, the impact of reverse transcription error in library preparation has not been thoroughly investigated.

A more direct way to measure mutation rates is to use a Luria-Delbrück fluctuation test (*Luria and Delbrück, 1943*; *Koziol, 1991*; *Foster, 2006*; *Furió et al., 2005*; *Combe and Sanjuán, 2014*). In this method, a large number of parallel cultures are infected with small inocula and assessed for a set of newly generated mutants exhibiting a scoreable phenotype after a period of exponential growth. Because the mutations are rare and random, they follow a Poisson distribution across cultures. Mutation rate estimates from a null class model are robust to the mode of replication, which may vary across viral species (*Foster, 2006*; *Furió et al., 2005*). Using resistance to monoclonal antibodies as a scoreable phenotype, influenza's mutation rate has been estimated to be $2.7 \times 10^{-6}$ to $3.0 \times 10^{-5}$ substitutions per nucleotide per strand copied (*Suárez et al., 1992*; *Suárez-López and Ortín, 1994*). While fluctuation tests are more precise than sequencing assays, most scoreable phenotypes sample just a few sites or mutational classes.

Here, we apply two new approaches for measuring the influenza virus mutation rate that overcome the drawbacks of sampling bias and low statistical power inherent to currently available methods. The first relies on measurements of the frequency of mutations to stop codons within a short segment of the influenza genome using PrimerID, an error-controlled next-generation sequencing approach (*Jabara et al., 2011*; *Zhou et al., 2015*). Because these nonsense mutations are lethal and generally not propagated, their frequencies approximate the mutation rate in the prior replication cycle (*Cuevas et al., 2009*). The second is a Luria-Delbrück fluctuation test that scores reversion to fluorescence in virally encoded green fluorescent protein (GFP) mutants (*Zhang et al., 2013*). The GFP method enabled interrogation of all 12 mutation classes independently and under distinct replication conditions.

## Results

The vast majority of premature stop codons in RNA virus open reading frames are lethal and are therefore likely to have been generated during the previous replication cycle (for example, [*Visher et al., 2016*]). Eighteen of the 61 sense codons are a single mutation away from a stop codon, and the frequency at which these nonsense mutational targets (NSMT) mutate to stop codons approximates the viral mutation rate. When combined with a highly accurate next generation sequencing approach, the NSMT method can provide rate estimates for eight mutational classes (*Cuevas et al., 2009*; *Cuevas et al., 2015*; *Acevedo et al., 2014*). We identified a 402 base fragment within the PA gene of A/Wisconsin/03/2007 H3N2 that contains a balanced distribution of 80

NSMT, and used the PrimerID method to sequence individual PA clones from an influenza population on the Illumina platform. PrimerID sequencing utilizes a library of barcoded reverse transcription primers to generate consensus sequences for each cDNA template, thereby controlling for the PCR or base-calling errors that plague many next generation sequencing studies (*Jabara et al., 2011*; *Zhou et al., 2015*). The PrimerID method does not control for errors introduced during reverse transcription (RT), and the mutation rates of reverse transcriptases are similar to those of viral RNA dependent RNA polymerases (RdRp) (*Sanjuán et al., 2010*).

In an attempt to distinguish RT errors from mutations introduced by the influenza RdRp, we compared PrimerID-NSMT estimates of the mutation rate for influenza virus to a control, in which the segment 3 (PA) RNA was expressed from a plasmid pol I promoter in transfected cells (*Hoffmann et al., 2000*). Mutations identified in the viral genome are derived from either the influenza RdRp or RT, and the control establishes the background error rate of the assay due to pol I transcription and RT (*Figure 1A*). We obtained over 449,000 aligned PA fragment consensus sequences for each sample, representing approximately 75% of starting RNA templates. The frequencies of mutations to stop codons were similar for 5 of the 8 mutation classes (*Figure 1B*, *Supplementary file 1*). The frequencies of the other three mutation classes (A to U, C to A, and U to G) were only slightly higher in the samples derived from RNA replicated by the influenza RdRp than those that were not. The G to A mutation rate was highest in both samples ($1.3 \times 10^{-4}$ and $8.5 \times 10^{-5}$ substitutions per nucleotide for cell-derived and viral-derived samples, respectively), and analyses of the RT mutational spectrum consistently show this to be the most common mutation, with rates of $1 \times 10^{-4}$ substitutions per nucleotide (*Gout et al., 2013*; *Mansky and Temin, 1995*; *Holtz and Mansky, 2013*; *Cuevas et al., 2015*). These data demonstrate that the background error rate of reverse transcriptase during sample preparation is equal to or higher than the rate of mutations introduced by the influenza RdRp.

We also compared the frequency of mutations to stop codons to the frequency of all observed mutations. Mutations in the transfected control were evenly distributed across the PA fragment and no more common than the subset of mutations to stop codons (*Figure 1C*). In contrast, the frequency of mutations in the replicated viral RNA was higher than the subset of stop codon mutations. The accumulation of mutations to frequencies above those of the stop codon mutations and the background signal indicate the action of selection on newly generated mutations. Together, these data suggest that the high background error rate of reverse transcriptase and issues of selection bias may confound sequencing-based measurements of the basal mutation rate and mutational bias of riboviruses.

## A fluorescence-based fluctuation test

We developed a Luria-Delbrück fluctuation test for influenza virus mutation rates that scores reversion to fluorescence in a set of 12 virally-encoded mutant green fluorescent proteins (GFP). The fluorescent chromophore of enhanced GFP contains three essential amino acids (T65, Y66, and G67) (*Ma et al., 2010*), and nonsynonymous substitution at any of these positions results in a GFP with either absent or altered fluorescent properties (*Timerghazin et al., 2008*; *Fu et al., 2015*; *Nakano et al., 2002*). We used a plasmid that contains GFP instead of hemagglutinin on influenza A virus segment 4 (ΔHA-GFP, [*Martínez-Sobrido et al., 2010*]), to generate a set of 12 derivatives that each encode a mutant GFP protein (*Table 1*). Each of these mutant GFP proteins has a single nucleotide mutation that, with reversion to fluorescence, will interrogate a specific mutational class introduced by the viral RdRp during viral replication. Influenza viruses expressing these mutant GFP proteins from a genomic context were rescued from cells co-transfected with each construct as well as 7 additional plasmids that express the remaining 7 genomic segments and the viral proteins encoded by each. Because these ΔHA-GFP viruses express GFP rather than HA, they were replicated in cells stably expressing the HA protein in trans. They were then transferred to a second plate of non-HA expressing cells for imaging.

Because our mutant GFP proteins are not fluorescent, we used anti-GFP antibody staining and immunofluorescence microscopy to verify GFP expression from each of the 12 mutant ΔHA-GFP viruses. In virally infected cultures, we occasionally identified rare cells expressing GFP that was fluorescent at the excitation and emission wavelengths consistent with reversion to fluorescence (*Figure 2A*). We used antibody staining to titrate the total number of viruses expressing GFP. The growth kinetics of mutant ΔHA-GFP A/Puerto Rico/8/1934 H1N1 viruses were slower than the

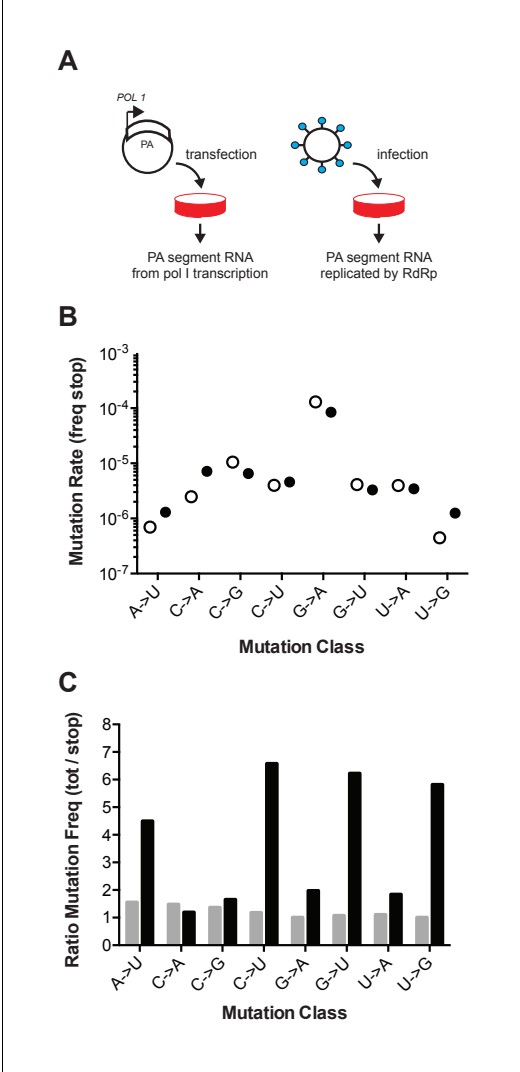

**Figure 1.** Influenza mutation rates by PrimerID next generation sequencing. (**A**) Segment 3 (PA) RNA was isolated from either cells transfected with pol I expression plasmids or cell-free supernatants of cells infected with influenza virus. These RNA were reverse transcribed with barcoded PrimerID primers and amplified by PCR for sequencing as described in the methods. We obtained 449,655 PrimerID consensus sequences for the plasmid-derived RNA sample and 481,286 consensus sequences for the virus-derived RNA sample. (**B**) The frequencies of mutations to stop codons in pol I transcribed RNA (open circles) and virus-derived RNA (filled circles) were determined by dividing the number of stop codon mutations across the consensus sequences by the total number of nonsense mutation target (NSMT) sites analyzed. Plotted data are in *Figure 1—source data 1*. See also *Supplementary file 1*. (**C**) Total mutation frequencies were calculated as the number of observed mutations for a particular mutation class divided by the number of sequenced sites that could mutate by that same class. Shown is the ratio of total mutation frequency to stop

*Figure 1 continued on next page*

parental PR8, but similar among the 12 mutants (*Figure 2B*). In all cases, titers of $1 \times 10^5$ per milliliter were achieved by 22 hr at 37°C. This corresponds to $10^4$ viruses per well of a 96 well plate, which is the maximum that can be accurately measured by fluorescence microscopy. In subsequent experiments, we used antibody staining of infected cells to titrate the total number of viruses expressing GFP – the mutational target – since a subset of viruses will delete the GFP open reading frame during replication.

Fluctuation tests are most accurate when the marker is selectively neutral (*Foster, 2006*; *Furió et al., 2005*). We measured the replicative fitness of viruses expressing the mutant ΔHA-GFP relative to those expressing the wild type ΔHA-GFP. We competed a subset of the mutant viruses against a wild type ΔHA-GFP virus containing a neutral PB1 sequence barcode and used RT-quantitative PCR to measure the frequency of the competitors over serial passage on MDCK-HA cells (*Visher et al., 2016*). Each of 6 mutant ΔHA-GFP viruses, which sample mutations in the 3 mutated amino acid positions, maintained stable frequencies over 4 passages. They were just as fit as the wild type ΔHA-GFP virus (*Figure 2B*, p>0.05, n = 3 replicates, one way ANOVA), confirming that the scoreable phenotype and the mutations interrogated are selectively neutral.

The secondary structure of genomic RNA in positive sense viruses is known to influence mutation rates in a site specific manner (*Geller et al., 2015*, *2016*; *Pathak and Temin, 1992*; *Pita et al., 2007*). In influenza virus, the formation of stable RNA structures in the replication complex is limited by the binding activity of the viral nucleoprotein (*Te Velthuis and Fodor, 2016*). We performed an in silico analysis of the ΔHA-GFP RNA to further exclude the possibility that reversion of mutations could be influenced by local RNA structure (*Figure 2D*). A sliding window analysis of the minimum free energy of RNA folding suggests that the introduced mutations are not located in highly stable RNA secondary structures in the ΔHA-GFP RNA (*Lorenz et al., 2011*; *Jorge et al., 2015*). The rates at which these mutations revert to the wild type sequences are therefore likely to be representative of mutation rates across the influenza virus genome.

We used a Luria-Delbrück fluctuation test to convert reversion frequencies to viral mutation rates (*Luria and Delbrück, 1943*; *Foster, 2006*). For each of the 12 mutant ΔHA-GFP, we infected parallel cultures of MDCK-HA cells with mutant ΔHA-GFP viruses and transferred replicated virus to MDCK cells in a 96-well imaging plate

*Figure 1 continued*

codon mutation frequency by mutation class for sequences derived from plasmid-derived RNA (grey bars) and virus-derived RNA (black bars). Plotted data are in *Figure 1—source data 2*.

The following source data is available for figure 1:

**Source data 1.** Spreadsheet with frequencies of mutations to stop codons in plasmid- and virus-derived RNA.
**Source data 2.** Spreadsheet with mutation frequency by class for plasmid- and virus-derived RNA.

(*Figure 3A*). The replication time and transfer volume were empirically determined for each mutation class, drug, temperature, and virus tested to yield approximately $10^4$ infectious viruses in the transfer population. Because the ΔHA-GFP viruses do not express the HA protein, they only replicate in MDCK-HA cells, and there was no viral spread in the imaging plate. The kinetics and intensity of GFP expression are driven by the ability of the replicated viruses to infect cells on the imaging plate, replicate their RNA and express GFP protein. Expression is independent of whether a given genome coding for a green GFP was generated early or late in the replication plate. We used a null class model to calculate mutation rates based on the number of parallel cultures without a revertant in the imaging plate (green fluorescence) and the degree of viral replication (anti-GFP antibody staining of the inocula and replicated virus). Importantly, the null class model relies on the number of cultures with any revertant virus, as opposed to the absolute number of revertant viruses. Because the actual number of infectious cycles is irrelevant to the calculation of mutation rates, the null class model further removes any bias attributable to mutations generated early or late in the replication plate. The mutation rates we report using this method are in the coding (+) sense of the RNA, rather than the genomic (-) sense.

We validated the specificity of our assay for specific mutational classes by performing a set of fluctuation tests in each of three different mutagenic nucleoside drugs. We and others have previously shown that ribavirin increases the frequency C to U and G to A transitions, 5-azacytidine increases the frequency of C to G and G to C transversions, and 5-fluorouracil increases the frequency of all transitions (A to G, C to U, G to A, and U to C) in influenza virus (*Pauly and Lauring, 2015*; *Cheung et al., 2014*). Each of these mutagens increased the rates of only the expected mutation classes (*Figure 3B–D*). In some cases, the amount of replicated virus was sufficiently low and the

**Table 1.** Non-fluorescent ΔHA-GFP constructs.

| Mutation Probed[*] | Nucleotide Sequece[†] | Amino acid Sequence[‡] |
|---|---|---|
| WT eGFP | acc uac ggc | T Y G |
| A -> C | a**A**a uac ggc | **K** Y G |
| A -> G | acc uac g**A**c | T Y **D** |
| A -> U | acc **A**ac ggc | T **N** G |
| C -> A | acc u**C**c ggc | T **S** G |
| C -> G | acc uac g**C**c | T Y **A** |
| C -> U | acc **C**ac ggc | T **H** G |
| G -> A | acc u**G**c ggc | T **C** G |
| G -> C | **uGg** uac ggc[§] | **W** Y G |
| G -> U | acc **G**ac ggc | T **D** G |
| U -> A | acc u**U**c ggc | T **F** G |
| U -> C | a**U**a uac ggc | **I** Y G |
| U -> G | acc uac g**U**c | T Y **V** |

[*]Mutations are in the mRNA coding sense.

[†]Nucleotides 193–201 of the eGFP reading frame are shown. Changes from wild type are in bold and italics. Site that allows reversion to fluorescence is capitalized.

[‡]Amino acids 65–67 of eGFP are shown. Changes from wild type are in bold and italics.

[§]This construct is able to revert to wild type GFP (S65).

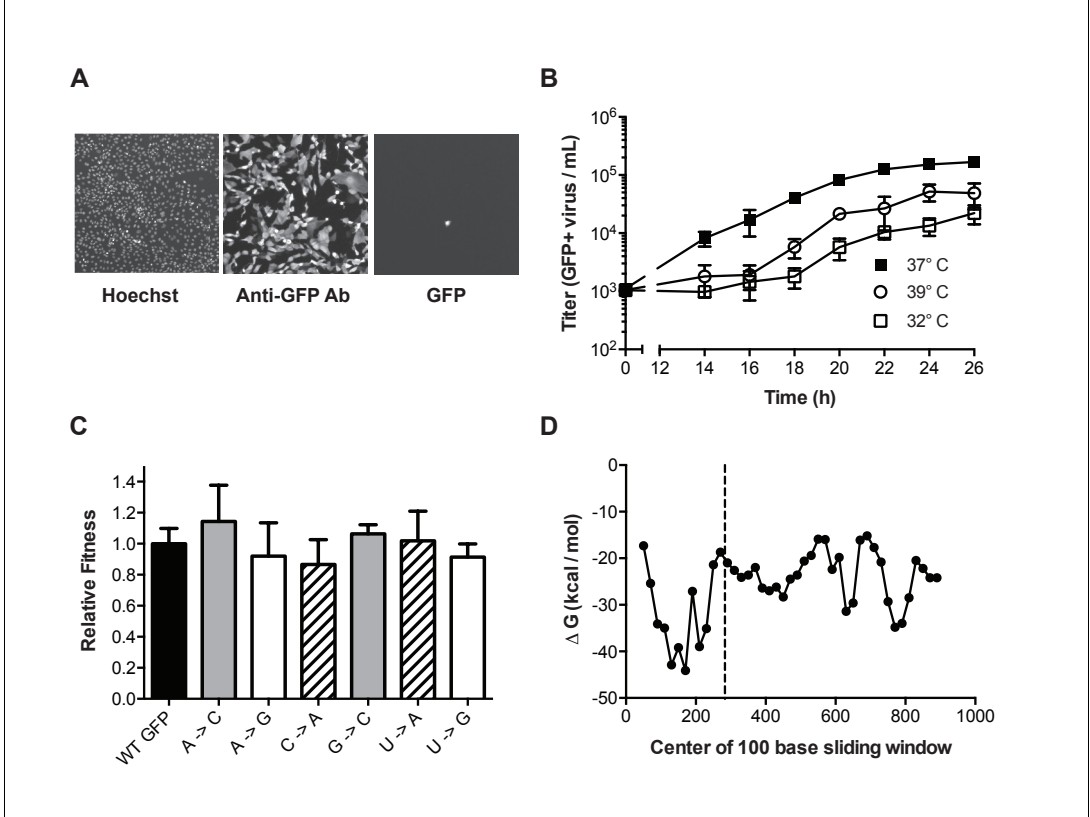

**Figure 2.** Characterization of mutant ΔHA-GFP influenza viruses. (A) Fluorescent images of cells infected with mutant ΔHA-GFP (shown are data for the A to C virus, see *Table 1*) and stained with Hoechst and anti-GFP Alexa 647 conjugate. Cells were imaged at 4x magnification and the resulting images were digitally magnified to an equal extent for this figure. (B) Growth kinetics of mutant ΔHA-GFP viruses. MDCK-HA cells were infected at an MOI of 0.01 in 96-well plates and incubated at 32°C (open squares), 37°C (filled squares), or 39°C (open circles). At each time point, the supernatants from 4 wells were transferred to a new 96-well plate containing MDCK cells. After 14 hr the cells were fixed and stained using an anti-GFP antibody. The number of cells stained were determined by fluorescence microscopy and used to calculate the titer of GFP expressing virus. Data shown are the cumulative mean and standard deviations for 4 measurements at each time point for each of two mutant ΔHA-GFP viruses (C to U and U to A viruses). Each point is the therefore the mean ± standard deviation for 8 values. Plotted data are in *Figure 2—source data 1*. (C) The fitness of 6 of the mutant ΔHA-GFP viruses (x-axis) were compared to wild type ΔHA-GFP through direct competition with a genetically barcoded competitor over 4 serial passages. Quantitative PCR was used to determine the relative changes in the frequency of the two competitors and fitness values were calculated as described in the Methods. Mutant viruses are classified by the GFP amino acid mutated, with wild type (black), T65 (gray), Y66 (striped), G67 (white). Shown are the mean and standard deviation for three competitions and fitness measurements for each virus. Plotted data are in *Figure 2—source data 2*. (D) The minimum free energy of RNA folding for 100 base sliding windows (80 base overlaps) were determined for the ΔHA-GFP construct. The location of the three mutated sites (bases 280–288) are indicated by the dashed line. Plotted data are in *Figure 2—source data 3*.

The following source data is available for figure 2:

**Source data 1.** Spreadsheet with virus titer (GFP+ virus per ml) in imaging plate at the indicated time points and temperatures.

**Source data 2.** Spreadsheet with replicate fitness values for wild type and ΔHA-GFP viruses as shown in *Figure 2C*.

**Source data 3.** Spreadsheet with minimum free energy of RNA folding by window start position as shown in *Figure 2D*.

mutations were sufficiently rare that we were not able to obtain replicate fluctuation tests in which the null class ($P_0$) lay within the ideal range of 0.1–0.7 (*Koziol, 1991*; *Foster, 2006*). Here and elsewhere, these less precise mutation rate measurements are indicated with open symbols (*Figures 3–5*).

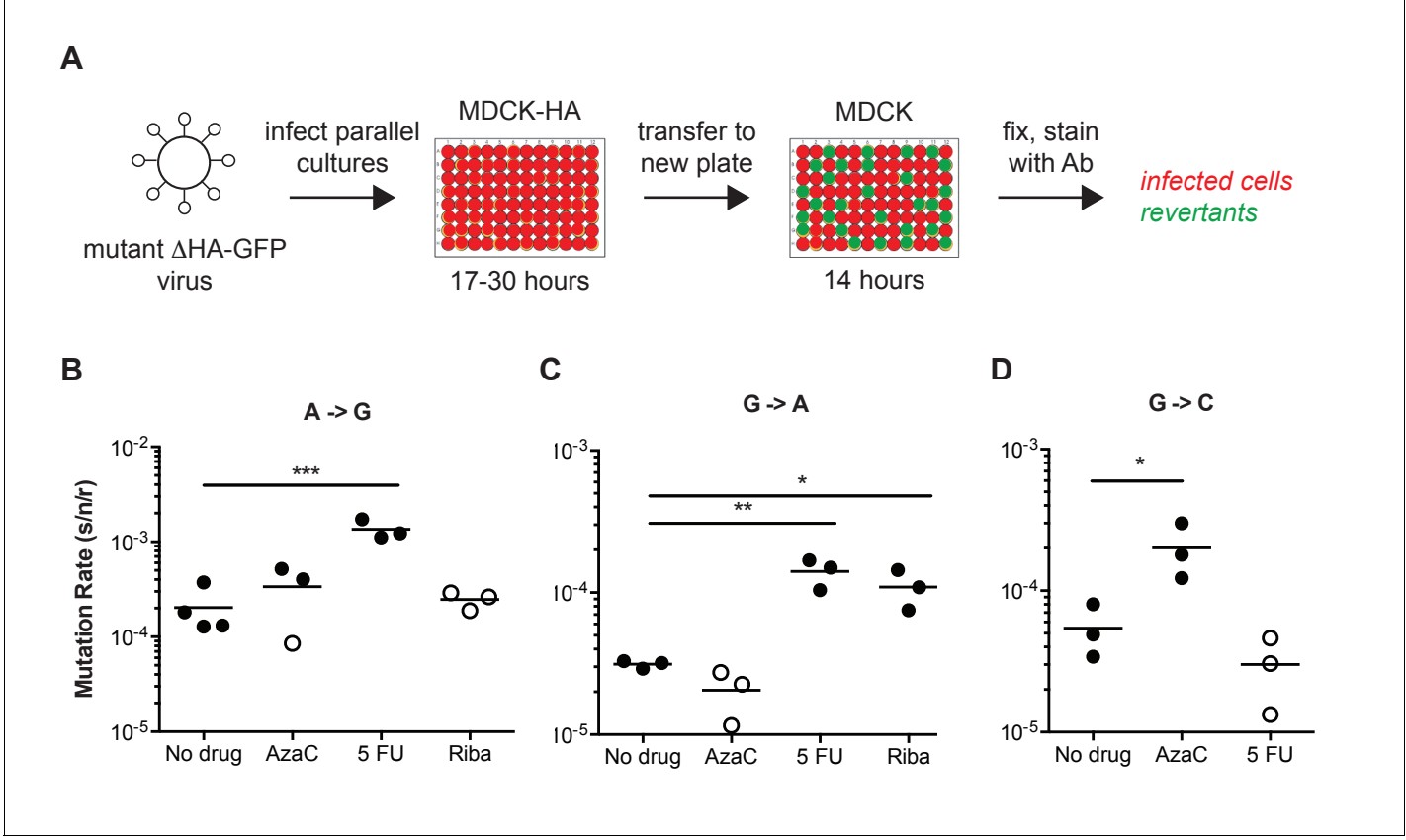

**Figure 3.** Fluorescent Luria-Delbruck fluctuation test. (**A**) General workflow for measuring the mutation rate using mutant *ΔHA-GFP* viruses. Parallel cultures of MDCK-HA cells were infected with passage one stocks of mutant *ΔHA-GFP* viruses at low multiplicity. The time for initial replication was varied to allow for a number of replicated viruses and revertants adequate to measure the mutation rate for a given class. Supernatants were transferred to 96-well plates of MDCK cells and incubated for 14 hr to allow for infection and GFP expression in target cells. The mutation rate for each mutant *ΔHA-GFP* virus and class was calculated as described in the methods and text based on the initial and final titer ($N_i$ and $N_f$, anti-GFP positive infected cells) and proportion of cultures with no revertants ($P_0$, wells without green fluorescence). (**B–D**) Specificity of the reversion to fluorescence assay. The (**B**) A to G, (**C**) G to A, and (**D**) G to C mutation rates for A/Puerto Rico/8/1934 H1N1 were measured at 37°C in cells pretreated with 0.625 μM 5-azacytidine (AzaC), 15 μM 5-fluorouracil (5 FU), or 2.5 μM ribavirin (Riba). No data are shown for G to C with 2.5 μM ribavirin because large titer decreases upon drug treatment prohibited measurements. Filled symbols represent measurements in which P0 is between 0.1 and 0.69, where the assay is most precise. Open circles represent data with P0 between 0.7 and 0.9. Arithmetic means are indicated. A one-way ANOVA with a Dunnett's correction for multiple comparisons was used for each mutation class to compare each drug treatment to no drug treatment. *p<0.05, **p<0.01, ***p<0.005. Plotted data are in *Figure 3—source data 1*.

The following source data is available for figure 3:

**Source data 1.** Mutation rates for A to G, G to A, and G to C viruses in the presence and absence of AzaC, 5FU and Riba as measured by fluctuation test.

## The mutation rates of influenza A virus

We used our GFP fluctuation test to measure the mutation rates of two evolutionarily divergent influenza viruses. Influenza A/Puerto Rico/8/1934 H1N1 (PR8) was the second influenza virus isolated and was extensively passaged in various cell culture environments prior to cloning (*Francis and Magill, 1935*). We cloned a circulating seasonal influenza virus, influenza A/Hong Kong/4801/2014 H3N2 (Hong Kong 2014), into the 8 plasmid reverse genetic system after limited passage in MDCK. After transfection, the rescued PR8 virus contained the *ΔHA-GFP* segment with seven PR8 genome segments. The rescued Hong Kong 2014 virus contained the *ΔHA-GFP* segment, the segments coding for the polymerase complex – PB2, PB1, PA, and NP – from A/Hong Kong/4801/2014 H3N2, and the segments encoding NA, M, and NS from PR8. This chimera was necessary to obtain high titer stocks.

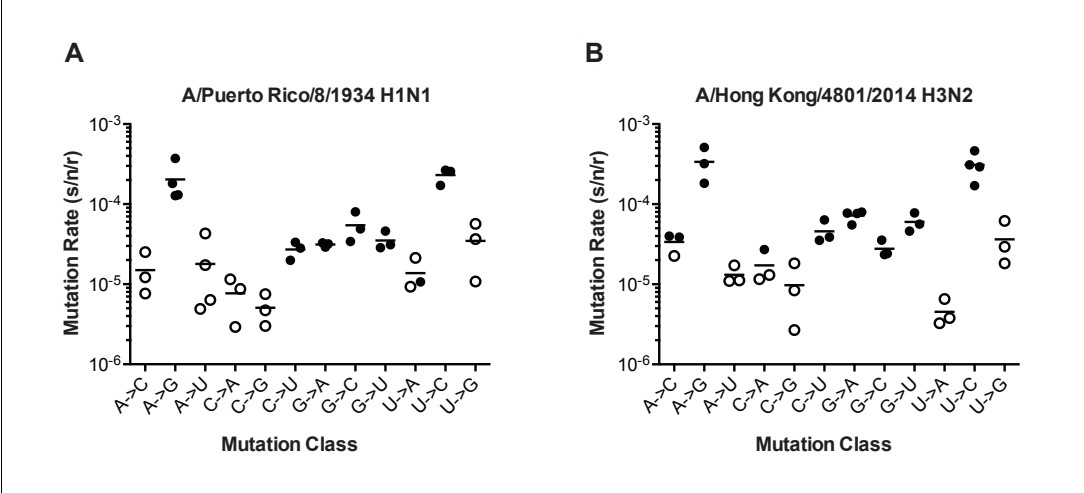

**Figure 4.** The mutation rates of influenza viruses replicated at 37°C. (**A**) Measurements of A/Puerto Rico/8/1934 H1N1 viruses encoding the 12 different mutant ΔHA-GFP constructs. (**B**) Measurements of viruses encoding the replication complex (PB2, PB1, PA, and NP) from A/Hong Kong/4801/2014 H3N2 and the remaining genes coming from A/Puerto Rico/8/1934 H1N1. Filled symbols represent measurements in which P0 is between 0.1 and 0.69. Open circles represent data with P0 between 0.7 and 0.95. Plotted data are in *Figure 4—source data 1*. Raw counts of green cells in positive wells for Hong Kong viruses are in *Figure 4—source data 2*. The arithmetic means are indicated on the graphs and the means and standard deviations reported in *Supplementary file 2*.

The following source data is available for figure 4:

**Source data 1.** Mutation rates for all twelve mutational classes for PR8 and Hong Kong viruses as measured by fluctuation test.
**Source data 2.** Raw data for all experiments with Hong Kong viruses showing number of wells with n green cells (n = 0–10).

The mutation rates of the PR8 and Hong Kong 2014 viruses were higher than previously reported for influenza A virus and generally biased toward transitions (*Parvin et al., 1986*; *Nobusawa and Sato, 2006*; *Bloom, 2014*). In both viruses, mutation rates were highest for the reciprocal transitions, A to G and U to C (*Figure 4* and *Supplementary file 2*). The rates for the other two transitions (C to U and G to A) were approximately six fold lower and similar to the rates of the more common transversion mutations. We note that this G to A mutation rate is much lower than the rate estimated using the PrimerID-NSMT assay (see *Figure 1*). This discrepancy may reflect differences in mutational bias between the influenza RdRp and retroviral reverse transcriptases. The overall rate and spectrum of mutations for the PR8 and Hong Kong 2014 viruses are very similar, and given the base composition of each virus, we estimate that each replicated 13.5 kb genome contains, on average, 2 to 3 mutations. The transition to transversion ratio is 2.7 in PR8 and 3.6 in Hong Kong 2014.

We did identify differences between the two viruses in specific mutation classes. The rate of G to A mutations was two-fold higher in Hong Kong 2014 than in PR8 ($7.2 \times 10^{-5}$ vs. $3.1 \times 10^{-5}$, p=0.0018, multiple t-test with Holm-Sidak correction), and the Hong Kong 2014 virus also exhibited a marginally increased rate of G to U mutations that was not statistically significant ($6.0 \times 10^{-5}$ vs. $3.5 \times 10^{-5}$, p=0.083). For both viruses, the rates of mutations away from A are symmetrical to the reciprocal mutations away from U. Interestingly, mutations away from C were much less common than the reciprocal mutations away from G. In PR8 G nucleotides are 3.8 times more likely to mutate than C nucleotides. In the Hong Kong virus, this difference is 2.7 fold.

## Influenza virus mutation rates across physiologic temperatures

Biochemical studies of purified influenza virus RdRp suggest that replication temperature can affect enzyme fidelity (*Aggarwal et al., 2010*). Influenza viruses replicate over a range of temperatures in nature from 32°C and 37°C across the respiratory tract of humans to 39°C in febrile illness to 41°C in birds (*Köhl, 1990*; *McFadden et al., 1985*; *Scull et al., 2009*; *Bradel-Tretheway et al., 2008*). We used the PR8 virus encoding mutant ΔHA-GFP representing the 5 most frequent mutational classes

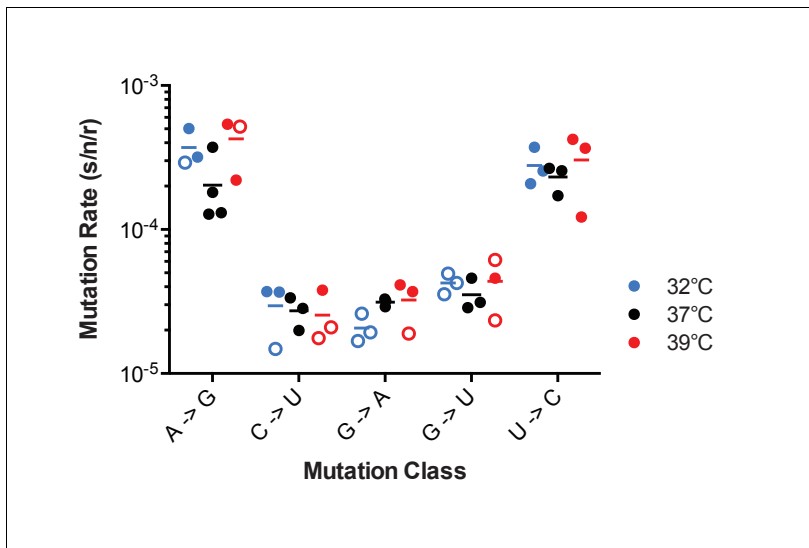

**Figure 5.** The effect of temperature on influenza A virus mutation rates. Mutation rates were determined for A/Puerto Rico/8/1934 H1N1 viruses encoding the indicated mutant ΔHA-GFP constructs replicated at 32°C (blue), 37°C (black) and 39°C (red). Filled symbols represent measurements in which P0 is between 0.1 and 0.69. Open circles represent data with P0 between 0.7 and 0.90. The arithmetic means are indicated. A two-way ANOVA revealed no significant differences in mutation rates based upon temperature. Plotted data are in *Figure 5—source data 1*.

The following source data is available for figure 5:

**Source data 1.** Mutation rates for all twelve mutational classes for PR8 at the indicated temperatures as measured by fluctuation test.

---

to measure mutation rates at different temperatures. This virus replicated reasonably well in MDCK-HA cells at 32°C and 39°C, albeit to lower titers (see *Figure 2C*). The mutation rates for these 5 classes were generally stable over this 7 degree range of physiological temperatures (*Figure 5*). We were unable to measure mutation rates at temperatures higher than 39°C due to host cell intolerance.

## Discussion

We developed two new methods to define the mutation rate and mutational bias of H1N1 and H3N2 influenza viruses. We found that the background error rate of reverse transcriptase may confound measurements of influenza virus mutation rates that are based on sequencing of RT-PCR amplified templates. We therefore developed a high throughput GFP-based assay to estimate the mutation rates for all 12 substitution classes. This assay can be easily adapted to any virus that tolerates the addition of the GFP open reading frame. While PR8 (H1N1) and Hong Kong 2014 (H3N2) viruses varied in their mutation rates for individual classes, the overall mutation rate was consistent across these evolutionarily divergent influenza polymerases at a range of temperatures. These mutation rates are considerably higher than previously reported, and given the impact of mutational load, suggest that the virus is replicating at the maximally tolerable mutation rates.

Sequencing assays for viral mutation rates are plagued by ascertainment and sampling biases. The mutations that are detected in plaque-derived populations represent only the viable fraction and those identified in passaged supernatants are often heavily biased toward mutations with less deleterious fitness effects. While Sanjuan and colleagues have appropriately adjusted for typical viral mutational fitness effects in sequence-based estimates of mutation rates (*Sanjuán et al., 2010*), these fitness effects may not be uniform across viruses or in the genes analyzed (*Sanjuán, 2010*; *Visher et al., 2016*). Next generation sequencing can minimize these biases by improving the detection of rarer, more deleterious mutations (*Geller et al., 2016, 2015*; *Cuevas et al., 2015*;

Combe et al., 2015). However, our data from PrimerID-controlled next generation sequencing suggest that reverse transcriptase error can be a significant confounder that needs to be considered in studies of RNA viruses. In our experiments, the high background RT error rate made it difficult to distinguish mutations introduced by the influenza polymerase complex from those generated during reverse transcription of the viral genomic RNA. The mutational bias of RT may also differ from that of viral RNA-dependent RNA polymerases. Guanine to adenine transitions are the most common mutation made by RT (Gout et al., 2013; Mansky and Temin, 1995; Holtz and Mansky, 2013; Cuevas et al., 2015) and are the ones found most frequently in our study as well as many others that rely on RT-PCR amplification for sequencing (for example, [Crotty et al., 2000; Pauly and Lauring, 2015; Cheung et al., 2014; Presloid et al., 2016]). In contrast, our fluctuation test suggests that A to G and U to C mutations are the most common classes in influenza.

Fluctuation tests are sensitive for rare mutational events and avoid many of the issues with sequencing assays (Luria and Delbrück, 1943; Foster, 2006; Furió et al., 2005; Combe and Sanjuán, 2014). Our reversion to fluorescence assay has several additional advantages over ones that rely on phenotypic markers such as drug or antibody resistance (Zhang et al., 2013). First, the marker was selectively neutral, as the mutant GFP and revertant wild type GFP viruses had equal fitness. Second, we were able to measure all 12 mutational classes in a format that allowed for sufficient replicates. Third, we were able to control the number of cellular infection cycles by expressing the HA protein in trans (Martínez-Sobrido et al., 2010). Fourth, we used an anti-GFP antibody to measure the number of mutation targets directly. One shortcoming of all fluctuation tests is that genomic mutation rates are extrapolated from data at one specific site. While RNA structures are unlikely to play a major role in mutation rate variability in influenza virus (Te Velthuis and Fodor, 2016), we cannot exclude that sequence context could modulate mutation rates across the genome. For example, influenza and other RNA viruses can exhibit bias in their dinucleotide content (Belalov and Lukashev, 2013; Greenbaum et al., 2008), and it is not clear whether the bases preceding a site can influence local nucleotide misincorporation rates.

We found that the mutation rates of the lab adapted PR8 H1N1 strain are similar to those of a recently circulating H3N2 strain, and both sets of measurements are considerably higher than those obtained in previous sequence-based studies. While these earlier works estimated rates between $7.1 \times 10^{-6}$ and $4.5 \times 10^{-5}$ mutations per nucleotide per cell infection, our composite mutation rates were $1.8 \times 10^{-4}$ and $2.5 \times 10^{-4}$ mutations per nucleotide per strand replicated for PR8 (H1N1) and Hong Kong/2014 (H3N2), respectively (Parvin et al., 1986; Nobusawa and Sato, 2006; Bloom, 2014; Sanjuán et al., 2010). Consistent with the biases detailed above, our measurements are closer to those obtained for specific classes in antibody-based fluctuation tests (Suárez et al., 1992; Suárez-López and Ortín, 1994). These very high mutation rates mean that each replicated genome has, on average, 2–3 mutations. We have found that 28–31% of randomly selected mutations in influenza virus are lethal (Visher et al., 2016). Using a 70% probability that a given mutation results in a viable virus, the likelihood of any given genome being able to replicate is only 34% to 49%. We suggest that mutational load accounts for a sizable portion of the 90–99% of genomes in influenza populations that are non-infectious (for example, [Baranovich et al., 2013; Pauly and Lauring, 2015]). This mutation rate clearly places influenza close to a theoretical maximum rate, and we and others have shown that small increases in the virus' mutation rate lead to considerable losses in genome infectivity (Baranovich et al., 2013; Cheung et al., 2014; Pauly and Lauring, 2015).

Cellular replication environments are often hypothesized to influence RNA virus mutation rates, and yet these effects have rarely been documented (Pita and Roossinck, 2013; Pita et al., 2007; Diamond et al., 2004; Holtz and Mansky, 2013; Combe and Sanjuán, 2014). Here, we found no significant differences in rates of the five most common mutational classes across a seven degree range of physiologic temperatures. Nucleotide pools could potentially influence the observed mutational biases. Intracellular concentrations of nucleotide triphosphates are much higher than those of deoxynucleotides, and cellular pools are typically biased towards ATP and GTP, which have other metabolic functions (Traut, 1994; Stridh, 1983). While it is tempting to speculate that pool bias could lead to the observed asymmetry in mutations away from guanine, this is unlikely to be the case in MDCK cells. The concentrations of all four NTP in MDCK cells are at least ten fold higher than the $K_m$ of the influenza polymerase for each (Aggarwal et al., 2010; Stridh, 1983; Zhang et al., 2010). We can't exclude that biases in pools could play a role in primary cells where NTPs may be more limiting. However, Combe and Sanjuan found that VSV mutation rates were

similar across a range of primary and immortalized cell lines that were cultured under a range of conditions (*Combe and Sanjuán, 2014*). It is also intriguing that A to G and U to C were the most common mutations, as these classes are characteristic of the host enzyme adenosine deaminase acting on RNA (ADAR) (*Samuel, 2011*; *Bass, 2002*; *tenOever et al., 2007*). As ADAR-editing occurs almost exclusively on double stranded RNA, it is not clear that it would contribute to the mutation rates measured on our presumably unstructured GFP messages.

We expect that our data will lead to improved models of influenza evolution. For example, our estimates of the virus' transition to transversion bias can inform null models for inference of selection in protein coding genes. The availability of a complete nucleotide substitution matrix will also enable studies of selection on codon usage and dinucleotide content. The nucleotide frequencies of both PR8 (H1N1) and Hong Kong 2014 (H3N2) are far from what would be predicted by the 12 mutation rates. This suggests either that selection is maintaining the virus' nucleotide content away from the mutational equilibrium or that the virus has not had sufficient time to achieve it. Finally, our measurements for the rate of each mutation class, coupled with recent studies on mutational fitness effects in influenza will also greatly improve our ability to construct more accurate phylogenies.

## Materials and methods

### Viruses, plasmids, and cells

Madin-Darby canine kidney (MDCK) cells were provided by Arnold S. Monto (University of Michigan School of Public Health) who obtained them directly from the ATCC and Influenza Reagent Resource (Manassas, VA), and HEK 293 T cells were provided by Raul Andino (University of California, San Francisco). Both cell lines were maintained in Dulbecco's modified Eagle medium (Gibco 11965, Waltham, MA) supplemented with 10% fetal bovine serum (Gibco 10437) and 25 mM HEPES (Gibco 15630). Cells were maintained at 37°C and 5% $CO_2$ in a humidified incubator except where indicated. Laboratory stocks of these cell lines tested negative for mycoplasma contamination in 2013. Neither cell line has been independently authenticated in our lab. We have not used cell lines from the list of commonly misidentified cell lines maintained by the International Cell Line Authentication Committee.

Influenza A/Puerto Rico/8/1934 H1N1 was obtained from the ATCC (VR-1469). The A/Hong Kong/4801/2014 H3N2 strain was obtained from the Centers for Disease Control and Prevention International Reagent Resource (FR-1483). The A/Wisconsin/03/2007 H3N2 strain was provided by Dr. Arnold S. Monto (University of Michigan School of Public Health). Molecular clones were derived from each of these isolates by reverse transcription polymerase chain reaction (RT-PCR) amplification and insertion of all eight genomic segments into the pHW2000 plasmid (*Hoffmann et al., 2001*, *2000*).

Cells expressing the hemagglutinin (HA) protein of influenza A/Puerto Rico/8/1934 H1N1 (MDCK-HA cells) were generated by co-transfection Madin Darby canine kidney (MDCK) cells with pCABSD, which expresses a gene for Blasticidin S resistance, and pCAGGS-HA, which expresses the influenza A/Puerto Rico/8/1934 H1N1 HA (*Martínez-Sobrido et al., 2010*). Pools of cells stably expressing HA were selected in growth media containing 5 µg/mL Blasticidin S. These pools were enriched for cells with high HA expression by staining with an anti-HA antibody (1:1000 dilution, Takara c179, Mountain View, CA) and an Alexa 488-conjugated anti-mouse IgG (1:200 dilution, Life Technologies A11001, Waltham, MA) followed by fluorescence-activated cell sorting on a FACSAria II (BD Biosciences, San Jose, CA). Cells were sorted three times over the course of 5 passages and >99% of cells in the final population were positive for high level HA expression.

A pPOLI vector encoding eGFP with influenza genomic packaging sequences was kindly provided by Luis Martinez-Sobrido (University of Rochester). This construct, which we call ΔHA-GFP, expresses eGFP flanked by the 78 3'-terminal bases (33 noncoding, 45 coding) and 125 5'-terminal bases (80 coding, 45 noncoding) of segment 4 from influenza A/WSN/33 H1N1. It lacks the HA translation initiation codon (*Martínez-Sobrido et al., 2010*). Twelve mutant ΔHA-GFP constructs (*Table 1*) were generated using the QuikChange II site-directed mutagenesis kit (Agilent Technologies 200523, Santa Clara, CA) with primers 5'- CTCGTGACCACCCTG<mutant sequence>GTGCAGTGCTTCAGC-3' and 5'- GCTGAAGCACTGCAC< rev comp mutant sequence>CAGGGTGGTCACGAG-3', where

mutant sequence corresponds to the sequences in *Table 1* and rev comp mutant sequence is the reverse complement of each.

A neutral genetic barcode was incorporated into the PB1 segment of A/Puerto Rico/8/1934 H1N1 in the pHW2000 vector by overlap extension PCR using inner primers 5'-gatcacaactcatttC-caACgGaaACgGAgGgtgagagacaat-3' and 5'-ATTGTCTCTCACCCTCCGTTTCCGTTGGAAATGAGTTGTGATC-3', and outer primers containing BsmB1 sites for cloning into the pHW2000 plasmid.

Recombinant viruses were rescued in 12-well plates after transfection of co-cultures of $2 \times 10^5$ 293 T cells and $1 \times 10^5$ MDCK cells with mixtures of pHW2000 plasmids encoding all 8 influenza genome segments (500 ng each) using 2 µL of TransIT-LT1 (Mirus 2300, Madison, WI) per nanogram of DNA (*Hoffmann et al., 2000*). Viruses expressing GFP were rescued in the same manner except that the pPOLI vector encoding ΔHA-GFP or its mutants and pCAGSS-HA were used in place of the pHW2000 plasmid encoding influenza HA, and MDCK-HA cells were used in place of MDCK cells.

## PrimerID sequencing

A custom R script (https://github.com/lauringlab/NGS_mutation_rate_assay) (*Pauly and Lauring, 2017*; copy archived at https://github.com/elifesciences-publications/NGS_mutation_rate_assay) was used to identify the 402 base region in the A/Wisconsin/03/2007 H3N2 genome (positions 865 to 1266 of the PA gene) with the highest concentration of nonsense mutational targets (NSMT). Total cellular RNA was isolated using Trizol (Life Technologies 15596) from 293 T cells 48 hr after transfection with a plasmid expressing A/Wisconsin/03/2007 H3N2 segment 3 (PA). Virus RNA was isolated using Trizol from cell free supernatants of MDCK cells infected with A/Wisconsin/03/2007 H3N2 virus at a multiplicity of infection (MOI) of 0.5 for 24 hr. In both cases, the RNA was treated with DNase I (Roche 04716728001, Indianapolis, IN) to remove residual plasmid DNA. The copy number of segment 3 (PA) RNA in each sample was determined by reverse transcription with SuperScript III (Invitrogen 18080051, Waltham, MA) and primer 5'-AGCAAAAGCAGG-3' followed by quantitative PCR on a 7500 Fast Real-Time PCR system (Applied Biosystems, Waltham, MA) with Power SYBR Green PCR Master Mix (Applied Biosystems 4367659) and primers 5'-TCTCCCATTTGTGTGGTTCA-3' and 5'-TGTGCAGCAATGGACGATTT-3'. A plasmid encoding PA was used to generate a standard curve to relate cycle threshold to copy number. The absence of plasmid DNA containing the PA sequence was confirmed by lack of signal in qPCR of RNA that was not reverse transcribed.

Sequencing libraries were prepared from $2 \times 10^5$ copies of segment 3 (PA) RNA using Accuscript high fidelity reverse transcriptase (Agilent Technologies 200820) and primer (5'-CCTACGGGAGG-CAGCAGNNNNNNNNNNAATTCCTCCTGATGGATGCT-3'), which binds to bases 842 to 861 of the PA gene (positive strand numbering) and contains a degenerate $N^{10}$ barcode sequence (1,048,576 unique sequences). Because the RNA copy number was just one-fifth of the total number of barcode sequences, it is unlikely that the same barcode would prime multiple complementary DNA (cDNA) molecules. Three separate reverse transcription reactions were performed for RNA harvested from both transfected and infected cells to increase the total number of RNA templates in the experiment. The resulting PrimerID barcoded cDNA was purified using Agencourt AMPure XP beads (Beckman Coulter A63881, Indianapolis, IN) to remove residual primers. The purified cDNA was amplified by PCR for 26 cycles (10 s at 98°C, 30 s at 69°C, and 30 s at 72°C) using Phusion high fidelity DNA polymerase (New England Biolabs M0530, Ipswich, MA) and primers 5'-CAAGCAGAA-GACGGCATACGAGAT < i7 > AGTCAGTCAGTATGGGGCTACGTCCTCTCCAA-3' and 5'-AATGATACGGCGACCACCGAGATCTACAC < i5 > TATGGTAATTGGCCTACGGGAGGCAGCAG-3', where i5 and i7 are 8 base Illumina indexing sequences. These primers contain the Illumina flow cell adapters at their 5'-ends. Unique index primers were used in the PCR for each of the three RT replicates. Products were gel purified using a GeneJET Gel extraction kit (Thermo Scientific K0691, Waltham, MA) and replicates were pooled with each product at 1.5 ng/µL. The two pooled sets (one for transfected cells and one for infected cells) were each sequenced on an Illumina MiSeq with $2 \times 250$ paired end reads, V2 chemistry, and the sequencing primers 5'-TATGGTAATTGGCCTACGGGAGG-CAGCAG-3', 5'-AGTCAGTCAGTATGGGGCTACGTCCTCTCCAA-3', and 5'-TTGGAGAGGACGTAGCCCCATACTGACTGACT-3'. Each pooled set, one derived from transfection and one from infection, made up half of the DNA input on a separate sequencing run with the remaining DNA being composed of bacterial genome libraries. This allowed for sufficient sequencing diversity at each base. We obtained over 15 million reads from each of the samples.

Consensus sequences that met empirically determined count cutoffs were generated for each PrimerID using Ruby scripts kindly provided by Ronald Swanstrom and colleagues (University of North Carolina). We obtained greater than 449,000 consensus sequences for each of the two samples, suggesting that nearly 75% of the original RNA templates ($6 \times 10^5$ copies among 3 separate reactions) were sampled. Consensus sequences were aligned to the A/Wisconsin/03/2007 H3N2 PA sequence using Bowtie2 and analyzed using Samtools. A custom Python script was used to determine the base composition at each position (https://github.com/lauringlab/NGS_mutation_rate_assay) (*Pauly and Lauring, 2017*) and the number of stop codons within each PrimerID consensus sequence. The mutation frequency for each of the eight mutational classes was determined by dividing the number of stop codons resulting from each class by the number of sites sequenced that could possibly mutate to a stop codon through that same class. Raw sequencing fastq files from this experiment are available at the Sequence Read Archive under BioProject accession number PRJNA347826.

## Competition assay

Equal quantities (TCID$_{50}$) of selected mutant ΔHA-GFP viruses were mixed with wild type ΔHA-GFP viruses containing a neutral sequence barcode in the PB1 gene, and used to infect $4 \times 10^5$ MDCK-HA cells in a 6-well plate at an MOI of 0.01. At 24 hr post infection, supernatants were harvested and infectious particles were titered by TCID$_{50}$. The resulting virus was passaged three more times on MDCK-HA cells, maintaining an MOI of 0.01 at each passage. Each viral competition was performed in triplicate. Viral RNA was harvested from the initial mixture and passaged supernatants using a Purelink Pro 96 viral DNA/RNA kit (Invitrogen 12280). Complementary DNA was synthesized using Superscript III and random hexamers. Quantitative PCR was used to determine the relative amount of total PB1 (primers 5'-CAGAAAGGGGAAGATGGACA-3' and 5'-GTCCACTCGTGTTTGC TGAA-3'), barcoded PB1 (primers 5'-ATTTCCAACGGAAACGGAGGG-3' and 5'-AAACCCCCTTA TTTGCATCC-3'), and non-barcoded PB1 (primers 5'-ATTTCCAACGGAAACGGAGGG-3' and 5'-AAACCCCCTTATTTGCATCC-3') in each sample. The relative amounts of barcoded and non-barcoded PB1 at each passage were normalized by subtracting the Ct threshold for the total PB1 primer set from the respective Ct thresholds (for example, $\Delta Ct = Ct_{competitior} - Ct_{total\ PB1}$). The normalized values at each passage were compared to the initial viral mixture to obtain a relative Ct ($\Delta\Delta Ct = \Delta Ct_{P1} - \Delta Ct_{P0}$). The relative Ct was converted to reflect the fold change in genome copies ($\Delta$ratio $= 2^{-\Delta\Delta Ct}$). The slope of the differences between the $\log_{10} \Delta$ ratios of the two viruses as a function of the passage number is equal to the $\log_{10}$ relative fitness of the non-barcoded virus ($[\log_{10}\Delta ratio_{non-barcoded} - \log_{10} \Delta ratio_{barcoded}]$/passage) (*Visher et al., 2016*).

## Growth curves

One hundred TCID$_{50}$ of each mutant ΔHA-GFP virus (in 100 μL of media) were used to infect $1.2 \times 10^4$ MDCK-HA in a 96-well plate. At two hour intervals between 14 and 26 hr post infection, supernatants from 4 wells were transferred to a black 96-well plate containing $1.5 \times 10^4$ MDCK cells and 50 μL of viral media. Virus equivalent to the initial inoculum was added to 4 wells so that the virus present at 0 hr post infection could be determined. At 14 hr after supernatant transfer, the cells were fixed, stained and imaged as described below.

## RNA minimum free energy

The minimum free energy of the ΔHA-GFP RNA was determined using the RNA sliding window python script that is included with the CodonShuffle package (*Jorge et al., 2015*).

## GFP-based Luria-Delbrück fluctuation test

Passage 1 (P1) stocks of ΔHA-GFP viruses were made by passing rescued virus once on MDCK-HA cells at an MOI of 0.01 for 48 hr. For each fluctuation test, 24 or more parallel cultures of MDCK-HA cells were infected with P1 influenza viruses encoding one of the twelve ΔHA-GFP mutants in viral media (Dulbecco's modified Eagle medium (Gibco 11965) supplemented with 0.187% BSA, 25 mM HEPES, and 2 μg/mL TPCK treated trypsin [Worthington Biochemical 3740, Lakewood, NJ]). Depending on the mutation class, these infections were performed in either 96-well plates ($1.2 \times 10^4$ cells infected with 400 TCID$_{50}$ of virus in 100 μL), 48-well plates ($3.6 \times 10^4$ cells infected with

1200 TCID$_{50}$ of virus in 300 µL), or 24-well plates (7.2 × 10$^4$ cells infected with 2400 TCID$_{50}$ of virus in 600 µL). At 17–30 hr post infection (depending on the mutation class, drug treatment, and assay temperature) supernatants were transferred to black 96-well plates (Perkin Elmer 6005182, Waltham, MA) containing 1.5 × 10$^4$ MDCK target cells and 50 µL of viral media. Supernatants from each well of 48-well and 24-well plates were transferred in 150 µL aliquots to 2 or 4 wells of the black 96-well plate, respectively. In addition to the supernatants derived from the parallel replication cultures, two to four wells were infected with the amount of virus used to seed these cultures (see N$_i$, below)

At 14 hr post-infection, cells were fixed with 2% formaldehyde for 20 min, rinsed with phosphate buffered saline (PBS), and permeabilized with 0.1% triton-X-100 for 8 min. Cells were then rinsed again with PBS, incubated at room temperature for one hour in PBS with 2% BSA and 0.1% tween-20 (PBS-T), and stained with 1:5000 Hoechst (Life Technologies 33342) and 1:400 anti-GFP Alexa 647 conjugate (Life Technologies A31852) diluted in 2% BSA in PBS-T for 1 hr. Cells were washed three times with PBS-T, and the plates were sealed with black tape prior to removal of the final wash. Plates were imaged using an ImageXpress Microscope (Molecular Devices, Sunnyvale, CA) using DAPI, Cy5, and FITC-specific filter cubes with a 4x magnification lens. Four non-overlapping quadrants were imaged from each well to ensure that the entire surface area was captured, Cellular nuclei and antibody stained cells were counted using MetaXpress version 6 software (Molecular Devices). Cells expressing fluorescent GFP were manually counted from the collected images.

Mutation rates were calculated using the null-class model, $\mu_{(s/n/r)}$ = -ln(P$_0$)/(N$_f$-N$_i$), where $\mu_{(s/n/r)}$ is the mutation rate per strand replicated, P$_0$ is the proportion of cultures that do not contain a cell infected by a virus encoding fluorescent GFP, and N$_f$ and N$_i$ are the final and initial viral population sizes, as determined by staining with the anti-GFP antibody, which recognizes both fluorescent and non-fluorescent eGFP (*Foster, 2006*; *Furió et al., 2005*). Cultures that contained a number of green cells greater than or equal to 0.8 (N$_f$/N$_i$) were removed from the calculation because they were likely to have contained a pre-existing fluorescent revertant in the inoculum. These events were extremely rare given the low titer inocula. The null class model is most precise when P$_0$ is between 0.1 and 0.7 (*Foster, 2006*). Due to the rarity of certain mutation classes and the constraints of the maximum viral population size per culture and per well on the imaging plate, not all of our measurements fell within this range. Measurements where the P$_0$ was above 0.7 are indicated in the graphical representations of our data.

Ribavirin (1-[(2$R$,3$R$,4$S$,5$R$)−3,4-dihydroxy-5-(hydroxymethyl)oxolan-2-yl]−1H-1,2,4-triazole-3-carboxamide) (Sigma-Aldrich R9644, St. Louis, MO) was dissolved in PBS at 100 mM. 5-azacytidine (4-Amino-1-($\beta$-D-ribofuranosyl)−1,3,5-triazin-2(1 hr)-one) (Sigma-Aldrich A2385) and 5-Fluorouracil (2,4-Dihydroxy-5-fluoropyrimidine) (Sigma-Aldrich F6627) were dissolved in dimethyl sulfoxide (DMSO) at 100 mM and 384 mM, respectively. For mutation rate measurements in the presence of drug, MDCK-HA cells were pretreated with viral media containing 2.5 µM ribavirin, 0.625 µM 5-azacytidine, or 15 µM 5-fluorouracil for three hours. Mutation rate assays were carried out according to the above protocol except that the viral media for the initial infections contained drugs at the indicated concentrations.

Mutation rate measurements at different temperatures were carried out as above, except that the initial replication was performed in incubators maintained at 32°C or 39°C. The imaging plates were maintained at 37°C for the 14 hr after the supernatant transfer.

The mutational transition to transversion ratio was calculated as sum of the rates of the 4 transition classes divided by the sum of the rates of the 8 transversion classes. This metric describes the relative likelihood of any new mutation being a transition as opposed to a tranversion.

## Acknowledgments

This work was supported by a Clinician Scientist Development Award from the Doris Duke Charitable Foundation (CSDA 2013105) and R01 AI118886, both to ASL. MDP was supported by the Michigan Predoctoral Training Program in Genetics (T32GM007544). We thank Judy Opp and April Cockburn from the microbial sequencing core of the University of Michigan Host Microbiome Initiative for assistance with next generation sequencing, JT McCrone for assistance with sequence analysis, and Nick Santoro from the University of Michigan Center for Chemical Genomics for assistance with high content imaging and analysis. We thank Robert Woods for helpful discussion.

## Additional information

### Funding

| Funder | Grant reference number | Author |
|---|---|---|
| National Institute of Allergy and Infectious Diseases | R01 AI118886 | Adam S Lauring |
| Doris Duke Charitable Foundation | CSDA 2013105 | Adam S Lauring |
| National Institute of General Medical Sciences | T32 GM007544 | Matthew D Pauly |

The funders had no role in study design, data collection and interpretation, or the decision to submit the work for publication.

### Author contributions

MDP, Conceptualization, Resources, Data curation, Software, Formal analysis, Validation, Investigation, Visualization, Methodology, Writing—original draft, Writing—review and editing; MCP, Data curation, Investigation, Methodology; ASL, Conceptualization, Data curation, Formal analysis, Supervision, Funding acquisition, Validation, Investigation, Visualization, Methodology, Writing—original draft, Project administration, Writing—review and editing

### Author ORCIDs

Adam S Lauring, http://orcid.org/0000-0003-2906-8335

## Additional files

### Supplementary files

• Supplementary file 1. Nonsense mutation counts from PrimerID sequencing of the influenza PA gene.

• Supplementary file 2. Influenza A virus mutation rates for PR8 and Hong Kong viruses.

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
