## [Decision Letter]

Thank you for submitting your article "A novel twelve class fluctuation test reveals higher than expected mutation rates for influenza A viruses" for consideration by *eLife*. Your article has been reviewed by three peer reviewers, one of whom, Karla Kirkegaard (Reviewer #1), is a member of our Board of Reviewing Editors and the evaluation has been overseen by Detlef Weigel as the Senior Editor. The following individual involved in review of your submission has agreed to reveal his identity: Claus O Wilke (Reviewer #3).

There are many recent papers that discuss the mechanisms and consequences of the high error rates of viral RNA-dependent RNA polymerases. This is one of the most thoughtful. The reported mutational frequencies and rates of mutation during RNA viral replication have been estimated previously by various methods, most fraught with assumptions that reduce the usefulness of the measurement.

Summary

In this manuscript, the error frequencies observed following plasmid-based expression with that of RNA replication-based expression were determined. Two approaches are described to determine and subsequently to remove any bias concerning fitness of mutated RNAs. In the first approach, only mutations that gave rise to stop codons by a single mutational event were assessed. Few differences were observed in the absence and presence of the function of the influenza virus polymerase. This attests to the well-known high error rate of reverse transcriptase, which confounds any analysis of previous mutational events. Here is where the authors should first mention the CirSeq approach of Acevado et al.,2014, which the present manuscripts cleverly illustrates is not sufficient for avoiding pressures of selection. Indeed, it is shown here that there is post-mutational selection for certain classes of mutations that are not dead-end mutations like the creation of a stop codon (Figure 1). This illustrates the necessity of not only removing RT error from calculated misincorporation, but also any selection pressure, including ongoing RNA replication within the cell after the mutational event has occurred.

The second approach relies upon a gene which does not provide a fitness advantage or disadvantage to the virus – eGFP. Although previous work assessed the mutation rates of influenza A virus as between 3 x 10-6 and 3 x 10-5, the present study results in a revised mutation rate of about 2 x 10-4. Consequently replication of the eight-segment genome should result in approximately three mutations per cycle of replication. In these experiments, selection is controlled by monitoring the gain of GFP fluorescence from modified viral genomes that encode GFP proteins that are one nucleotide away from function. Care is taken in Figure 2 to ensure that none of the mutations adds a fitness cost to the survival of the viral construct. These great experiments form the heart of the paper. Unfortunately, the reader is not provided with enough information to understand them well. At least, a curious but non-specialized reader cannot, and needs more description of the virological methods, statements of assumptions, and tests to see if those assumptions are valid, as described below.

The reviewers have discussed the reviews with one another and the Reviewing Editor has drafted this decision to help you prepare a revised submission. We hope you will be able to submit the revised version within two months. Please address the questions below in the revised manuscript.

Essential revisions:

1) All reviewers agree that there is insufficient evidence presented to distinguish between a 'stamping' model and others with a larger number of replicative events. The Discussion that begins in subsection “The mutation rates of influenza A virus” paragraph four is not backed up by data that the reviewer can access, and the conclusions of stamping vs. binary replication modes are sufficiently important that, unless more data are provided, this section should be deleted.

2) The virology presented here is complicated and well-designed. *eLife* has a wide audience that would benefit from more detailed explanation about the multiple-plasmid transfection to recover virus, the DeltaHA construct that then needs to be passaged in an HA-expressing line, etc. These details are in the Materials and methods, which is fine, but there are points that are relevant for the interpretation of the data. After transfection, for example, how long does it take for the first infectious cycle to complete? Is the 24-hour time point one such cycle?

3) In Figure 2, which mutant DeltaHA viruses are analyzed? Are the error bars because it is all mutant viruses, and is wild-type DeltaHA included? This experiment needs to be clarified.

4) Subsection “A fluorescence-based fluctuation test” The phrase "high minimum free energy" will confuse some non-physical chemists. Perhaps 'suggests that the mutations are not located in highly stable RNA secondary structures' would be more helpful for making the point.

5) Several improvements in clarity will improve the accessibility of the manuscript. For example, the 12 mutational classes are never defined. It would be good to do so in a few words (e.g., "all possible different single-nucleotide mutations"). The terminology of "missense" and "nonsense" mutations, while conventional, implies value statements. Greater clarity would be provided by "non-synonymous" and "mutation to stop codon". Note that the manuscript is somewhat inconsistent and uses both.

---

## [Author Response]

*[…] Essential revisions:*

*1) All reviewers agree that there is insufficient evidence presented to distinguish between a 'stamping' model and others with a larger number of replicative events. The Discussion that begins in subsection “The mutation rates of influenza A virus” paragraph four is not backed up by data that the reviewer can access, and the conclusions of stamping vs. binary replication modes are sufficiently important that, unless more data are provided, this section should be deleted.*

We recognize that our inference of linear vs. binary mode of replication relies on certain assumptions and concede that the evidence may be insufficient as presented. We also feel that the novelty of the manuscript lies elsewhere. Therefore, we have deleted the section of the Results and Discussion that deal with this question.

*2) The virology presented here is complicated and well-designed. eLife has a wide audience that would benefit from more detailed explanation about the multiple-plasmid transfection to recover virus, the DeltaHA construct that then needs to be passaged in an HA-expressing line, etc. These details are in the Materials and methods, which is fine, but there are points that are relevant for the interpretation of the data. After transfection, for example, how long does it take for the first infectious cycle to complete? Is the 24-hour time point one such cycle?*

We have added additional text in the Results, subsection “A fluorescence-based fluctuation test” to clarify how we used the 8 plasmid reverse genetics system to rescue recombinant viruses expressing GFP from transfected cells. We also clarified how these GFP viruses are only propagated on MDCK cells expressing HA in trans and how the use of non-HA expressing MDCK in the imaging plate allowed for infection, but not subsequent spread of GFP viruses.

With respect to the timing of the infectious cycle, we should be clear that none of the fluctuation tests were initiated with transfection. We used transfection of plasmids to rescue recombinant virus. This recombinant virus is a passage 0 (P0) stock. We then used this to infect cells and make a passage 1 (P1) stock, which was used for experiments. We usually harvest our P0 stocks ~ 48 hours after transfection. There is likely some degree of viral replication that takes place in this time period, but we have never quantified it.

We know from one step growth curves that the infectious cycle of PR8 is approximately 12-14 hours. Not surprisingly, the GFP virus grows slower (~17 hours), see for example Figure 2. As indicated in the text, the timing of transfer from the replication plate to the imaging plate was determined empirically for each GFP virus (17-24 hours at 37˚C) to ensure an adequate Nf population size (both for adequate fluorescent imaging and to target a P_0_ in the 0.1-0.7 range). Importantly, the number of infectious cycles on the replication plate is irrelevant to calculation of mutation rates using the null class model for fluctuation tests. The key parameters with respect to growth are the starting population (Ni) and the post-replication population size (Nf).

The imaging plate was fixed at 14 hours post infection. This time point was chosen as it was sufficient to allow for infection and expression of GFP from the genome of the transferred viruses used to infect the plate.

We have clarified these elements of experimental design in the Results section as above and in subsection “The mutation rates of influenza A virus”.

*3) In Figure 2, which mutant DeltaHA viruses are analyzed? Are the error bars because it is all mutant viruses, and is wild-type DeltaHA included? This experiment needs to be clarified.*

This experiment had 4 replicates of the C-U mutant ΔHA-GFP virus and 4 replicates of the U-A mutant ΔHA-GFP viruses virus in the PR8 genetic backbone. The error bars are standard deviation for 8 replicates. We have clarified this in the figure legend “Data shown are the cumulative mean and standard deviations for 4 measurements at each time point for each of two mutant ΔHA-GFP viruses (C to U and U to A viruses). Each point is the therefore the mean ± standard deviation for 8 values.” We note that the fitness values derived from competition assay for 6 mutant ΔHA-GFP viruses are a better measurement of growth relative to the “wild type” ΔHA-GFP virus.

*4) Subsection “A fluorescence-based fluctuation test” The phrase "high minimum free energy" will confuse some non-physical chemists. Perhaps 'suggests that the mutations are not located in highly stable RNA secondary structures' would be more helpful for making the point.*

This is a good point. We have changed the wording as suggested, subsection “A fluorescence-based fluctuation test”.

*5) Several improvements in clarity will improve the accessibility of the manuscript. For example, the 12 mutational classes are never defined. It would be good to do so in a few words (e.g., "all possible different single-nucleotide mutations"). The terminology of "missense" and "nonsense" mutations, while conventional, implies value statements. Greater clarity would be provided by "non-synonymous" and "mutation to stop codon". Note that the manuscript is somewhat inconsistent and uses both.*

We have changed “12 mutational classes” to “all possible single nucleotide mutations” and “the twelve different mutation types” in the Abstract. We then define mutational classes more clearly at the first instance in the Introduction.

We have changed most of the “nonsense” references to “stop codons.” We have kept the term “nonsense mutational targets (NSMT)” as it has been used in the mutation rate literature and is more concise than “codons that are one mutation away from a stop codon.”

We verified that we were similarly consistent with non-synonymous and avoided the use “missense” throughout.